# Diffusion Model with Cross Attention as an Inductive Bias for Disentanglement

**Tao Yang**[1][*], **Cuiling Lan**[2][†], **Yan Lu**[2], **Nanning Zheng**[1][†]

yt14212@stu.xjtu.edu.cn,
{culan, yanlu}@microsoft.com,
nnzheng@mail.xjtu.edu.cn

[1]National Key Laboratory of Human-Machine Hybrid Augmented Intelligence,
National Engineering Research Center for Visual Information and Applications,
and Institute of Artificial Intelligence and Robotics, Xi'an Jiaotong University, China
[2]Microsoft Research Asia

https://github.com/thomasmry/EncDiff

## Abstract

Disentangled representation learning strives to extract the intrinsic factors within the observed data. Factoring these representations in an unsupervised manner is notably challenging and usually requires tailored loss functions or specific structural designs. In this paper, we introduce a new perspective and framework, demonstrating that diffusion models with cross-attention itself can serve as a powerful inductive bias to facilitate the learning of disentangled representations. We propose to encode an image into a set of concept tokens and treat them as the condition of the latent diffusion model for image reconstruction, where cross attention over the concept tokens is used to bridge the encoder and the U-Net of the diffusion model. We analyze that the diffusion process inherently possesses the time-varying information bottlenecks. Such information bottlenecks and cross attention act as strong inductive biases for promoting disentanglement. Without any regularization term in the loss function, this framework achieves superior disentanglement performance on the benchmark datasets, surpassing all previous methods with intricate designs. We have conducted comprehensive ablation studies and visualization analyses, shedding a light on the functioning of this model. We anticipate that our findings will inspire more investigation on exploring diffusion model for disentangled representation learning towards more sophisticated data analysis and understanding.

## 1 Introduction

Disentangled representation learning aims to uncover and understand the underlying causal factors of observed data [1, 13]. This is believed to have immense potential to empower a multitude of machine learning tasks, facilitating the models to attain better interpretability, superior generalizability, controlled generation, and robustness [33]. Over the years, the field of disentangled representation learning has attracted significant academic interest and many research contributions. Numerous methods, encompassing Variational Autoencoders (VAE) based techniques (such as $\beta$-VAE [14, 2], FactorVAE [19]), Generative Adversarial Networks (GAN) based approaches (such as InfoGAN [5], InfoGAN-CR [23]), along with others [37, 28], have been proposed to advance this field further.

---

[*]This work was done during internship at Microsoft Research Asia.
[†]Corresponding authors.

38th Conference on Neural Information Processing Systems (NeurIPS 2024).

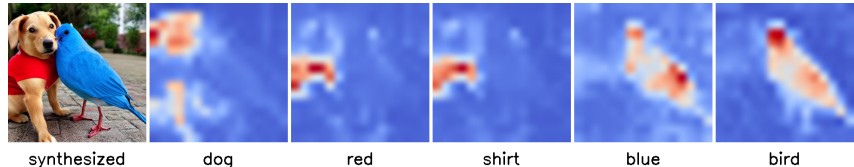

| synthesized | dog | red | shirt | blue | bird |

Figure 1: Average attention map across all time steps in stable diffusion. We draw inspiration from the process of text-to-image generation using a diffusion model with cross-attention. Utilizing the highly 'disentangled' words as the condition for image generation, the cross-attention maps observed from the diffusion model exhibit a strong text semantic and spatial alignment, indicating the model is capable of incorporating each individual word into the generation process for a final semantic aligned generation. This leads us to question whether such a diffusion structure could be inductive to disentangled representation learning.

Originally, Variational Autoencoders (VAEs) are designed as deep generative probabilistic models, primarily focusing on image generation tasks [20]. The core idea behind VAEs is to model data distributions from the perspective of maximizing likelihood using variational inference. Subsequent research has revealed that VAEs possess the potential to learn disentangled representations with appropriate regularizations on simple datasets. To enhance the disentanglement capability, a range of regularization losses have been proposed and integrated within the VAE framework [14, 19, 21]. Similarly, GANs have incorporated regularizations to promote the learning of disentangled features [5, 23, 44]. Despite significant progress, the disentanglement capabilities of these models remain less than satisfactory, and the disentangled representation learning is still very challenging. Locatello *et al*. demonstrate that relying solely on regularizations is insufficient for achieving disentanglement [24]. They emphasize the necessity of inductive biases from both the models and the data for effective disentanglement. A fresh perspective is eagerly anticipated to shed light on this field.

Recently, diffusion models have surfaced as compelling generative models known for their high sample quality [36]. Drawing inspiration from the evolution of VAE-based disentanglement methods, we are intrigued by the question of whether diffusion models, also fundamentally designed as deep generative probabilistic models, possess the potential to learn disentangled representations. Obtaining a compact and disentangled representation for a given image from diffusion models is non-trivial. Diffusion Autoencoder (Diff-AE) [26] and PDAE [42] advance the use of diffusion models for representation learning by encoding the image into a feature vector and incorporating this into the diffusion generation process. However, these representations do not exhibit disentanglement characteristics. What inductive biases are essential for the learning of disentangled representations? Could we have a diffusion-based framework possessing such inductive biases?

Notably, in text-to-image generation, a conditional diffusion model integrates the '*disentangled*' text tokens through cross attention, demonstrating the ability to generate semantically aligned images [30, 36, 12]. Interestingly, the observed cross-attention map reveals that different words have their corresponding spatial regions of high affinities, exhibiting strong semantic and spatial alignment as illustrated in Figure 1. These disentangled representations of 'words' could potentially contribute to a more streamlined generation process. Inspired by this, we wonder whether such diffusion structure with cross-attention can act as an inductive bias to facilitate disentangled representation learning.

In this paper, we investigate this question and explore the potential of diffusion models in disentangled representation learning. We discover that the diffusion model with cross attention can serve as a strong inductive bias to drive disentangled representation learning, even without any additional regularization. As illustrated in Figure 2 (a), we employ an encoder to transform an image into a set of concept tokens, which we treat as 'word' tokens, acting as the conditional input to the latent diffusion model with cross-attention. Here, cross attention bridges the interaction between the diffusion network and the image encoder. We refer to this scheme as EncDiff. EncDiff is powered by two valuable inductive biases, *i.e.*, information bottleneck in diffusion, and cross attention for fostering 'word' (concept token) and spatial alignment, contributing to the disentanglement. Experimental results on benchmark datasets demonstrate that EncDiff achieves excellent disentanglement performance, surpassing the previous methods with elaborate designs. Comprehensive ablation studies show that the strong disentanglement capability is mainly attributed to 1) the diffusion modelling and 2) the cross-attention interaction. Visualization analysis provides insights into the effectiveness of the disentangled representations.

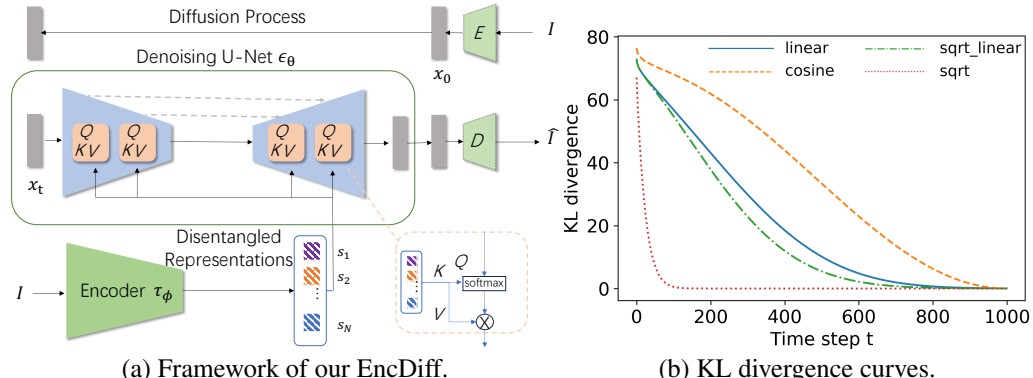

(a) Framework of our EncDiff.  (b) KL divergence curves.

Figure 2: (a) Illustration of our framework EncDiff. We employ an image encoder $\tau_\phi$ to transform an image $I$ into a set of disentangled representations, which we treat them as the conditional input to the latent diffusion model with cross attention. Here cross attention bridges the interaction between the diffusion network and the image encoder. For simplicity, we only briefly show the diffusion model which consists of an encoder $E$, a denoising U-Net and a decoder $D$ that reconstructs the image from the latent $x_t$. (b) Information bottleneck reflected by KL divergence in reverse diffusion process. The KL divergence between the data distribution $q(x_{t-1}|x_t, x_0)$ and the Gaussian prior distribution $\mathcal{N}(0, \mathbf{I})$ under four different variance ($\beta$) schedules: cosine, linear, sqrt linear and sqrt. The results have been normalized by the number of dimensions.

We have four main contributions.

- We reveal that the diffusion model with cross attention can act as a strong inductive bias for enabling disentangled representation learning.
- We introduce a simple yet effective framework, EncDiff, powered by a latent diffusion model with cross attention and an ordinary image encoder, for disentangled representation learning.
- Our framework inherently leverages two valuable inductive biases: the information bottleneck in diffusion, and cross attention, fostering concept token learning and spatial alignment. We analyze that the diffusion process inherently exhibits time-varying information bottlenecks, which is vital for disentangled representation learning.
- Without additional regularization or specific designs, our framework achieves state-of-the-art disentanglement performance, even outperforming the latest methods with more complex designs.

We anticipate that this new perspective will illuminate the field of disentanglement and inspire deeper investigations, paving the way for future sophisticated data analysis, understanding, and generation.

## 2  Related Work

**Disentangled Representation Learning** Disentangled Representation Learning endeavours to train a model proficient in disentangling the underlying factors of observed data [1, 13, 33]. A plethora of methods have been proposed to augment the generative models of VAEs and GANs, endowing them with disentanglement capability. These methods primarily rely on probability-based regularizations applied to the latent space. To improve disentanglement, most approaches focus on how to regularize the original VAE. For instance, this includes weighting the evidence lower bound (ELBO) as in $\beta$-VAE [14], or introducing different terms to the ELBO, such as mutual information (InfoVAE [43], InfoMax-VAE [29]), total correlation (Factor-VAE [19]), and covariance (DIP-VAE [21]). Locatello *et al.* [24] demonstrate that relying solely on these regularizations is insufficient for achieving disentanglement. DIP-VAE [21] introduces a regularizer on the expectation of the approximate posterior over observed data, by matching the moments of the distributions of latents. In this paper, we investigate the disentanglement capability of diffusion models and demonstrate that diffusion models with cross-attention can serve as a powerful inductive bias for disentanglement.

**Diffusion Models** Diffusion models have emerged as a powerful new family of deep probabilistic generative models [36], surpassing VAEs and GANs for image generation and many other tasks. Diffusion models progressively perturb data by injecting noise and then learn to reverse this process for the generation. A question arises as to whether diffusion models can effectively serve as disentangled

representation learners. It is challenging to obtain a compact yet disentangled representation of an image from a diffusion model. Diff-AE [26] and PDAE [42] investigate the possibility of using DPMs for representation learning, whereby an input image is autoencoded into a latent vector. However, these representations do not manifest disentangled characteristics. SlotDiffusion [35] and LSD [17] integrate diffusion models into object-centric learning, where diffusion acts as an improved slot-to-image decoder, and slot attention is still the key to promoting object-centric learning. Moreover, they aim to learn object-wise representation but still cannot disentangle the factors/attributes of an object/a scene. Limited research has explored disentangled representation learning by leveraging diffusion models. InfoDiffusion [34] encourages the disentanglement of the latent feature of Diff-AE [26] by introducing mutual information and prior regularization, similar to InfoVAE [43]. CL-Dis [18] introduces a VAE to guide the diffusion model to learn disentangled representation. DisDiff [40] employs a pre-trained diffusion model for disentangled feature learning. DisDiff adopts an encoder to learn the disentangled representations and a decoder to learn the sub-gradient field for each disentangled factor. It requires multiple decoders to predict these sub-gradient fields for all the factors and complex disentanglement losses, resulting in a costly and intricate process. The concurrent work SODA [16] leverages a diffusion model to generate novel view of an image for representation learning, revealing the capability of capturing visual semantics to diffusion model. Is it necessary to impose these complicated regularizations upon diffusion models as [40]? Does a strong inductive bias that facilitates disentanglement already exist within diffusion models? In this paper, we endeavour to answer these questions and illustrate that a simple framework driven by a diffusion model without any additional regularization is capable of achieving superior disentanglement performance.

## 3 Method

We aim to investigate the potential of diffusion models in disentangled representation learning. We propose a simple yet effective framework, EncDiff, that exhibits strong disentanglement capabilities, even without additional regularizations. We analyze and identify two valuable inductive biases, *i.e.*, information bottleneck in diffusion, and the cross attention for fostering 'word' (concept token) and spatial alignment, thereby promoting disentanglement. We elaborate on the framework design in subsection 3.1 and the analysis in subsection 3.2, respectively.

### 3.1 Framework of EncDiff

Figure 2 (a) illustrates the flowchart. It consists of an image encoder that transforms an input image into a set of concept tokens and a diffusion model that serves as the decoder to reconstruct the image. Cross-attention is employed as the bridge for the diffusion network and the image encoder.

**Image Encoder** For a given input image $I$, the image encoder $\tau_\phi$ aims to provide a set of concept tokens $\mathcal{S} = \{s_1, \cdots, s_N\}$, which act similarly to the word embeddings in the prompts for text-to-image generation in the latent diffusion models (LDMs) [30]. Without loss of generality, we use an ordinary CNN network as the image encoder to obtain concept tokens. Following the design of the encoder in VAE [20], we use a fully connected layer to transform the feature map into a feature vector. We treat each dimension of the encoded feature vector as a disentangled factor and map each factor to a vector (*i.e.*, concept token) by non-shared MLP layers (as illustrated in Figure 3).

**Diffusion Model with Cross Attention** We follow LDM [30] to construct our diffusion model in the latent space, which demonstrates superior generation ability. LDM is one of the most popular diffusion models, proposing to conduct diffusion denoising in the latent space. To condition the concept tokens during image generation, cross-attention is used to map these tokens into the intermediate representations of the U-Net in the diffusion model. This is accomplished by using the cross-attention defined as $Attention(Q, K, V) = \text{softmax}\left(\frac{QK^T}{\sqrt{d}}\right) \cdot V$, where the spatial feature in the intermediate feature map in diffusion model serves as a query, the concept tokens act as keys and values.

**End-to-End Training** We conduct an end-to-end training of the encoder and the diffusion model, utilizing the optimization objective of reconstructing noise, which is a methodology aligned with that employed in LDM [30].

## 3.2 Inductive Biases

We analyze that there are two crucial inductive biases in diffusion models: the information bottleneck in diffusion, and the cross-attention interaction. We analyze that the diffusion process inherently possesses the time-varying information bottlenecks.

### 3.2.1 Information Bottleneck in Diffusion

$\beta$-VAE [14] and AnnealVAE [2] leverage the Kullback–Leibler (KL) divergence in VAEs to enhance the disentanglement capability, where the KL divergence constraint plays a role of an information bottleneck. Here, we analyze the presence of an information bottleneck mechanism that promotes the disentanglement in diffusion models. Without loss of generality, our analysis focuses on the diffusion model in image latent space [30]. The analysis also holds in pixel space.

Within the framework of the latent diffusion model, we add Gaussian noise to an image latent $x_0$ over $T$ steps according to a variance schedule $\beta_1, \cdots, \beta_T$. This process yields a sequence of noisy samples $x_1, \cdots, x_T$,

$$q(x_t|x_{t-1}) := \mathcal{N}(x_t; \sqrt{1-\beta_t}x_{t-1}, \beta_t\mathbf{I}). \tag{1}$$

Let $\alpha_t = 1 - \beta_t$ and $\bar{\alpha}_t = \prod_{i=1}^t \alpha_t$. $x_t$ can be obtained using the following equation [15, 36]:

$$x_t = \sqrt{\bar{\alpha}_t}x_0 + \sqrt{1-\bar{\alpha}_t}\epsilon, \tag{2}$$

where the noise $\epsilon$ is sampled from a Gaussian distribution $\mathcal{N}(0, \mathbf{I})$.

The diffusion model optimizes a network (*e.g.*, U-Net) $\epsilon_\theta$ to predict the noise from the noisy input $x_t$ and the conditioning input $\mathcal{S}$ (concept tokens), with the loss function defined as

$$\mathcal{L}_r = \mathbb{E}_{x_0, \epsilon, t}\|\epsilon_\theta(x_t, t, \mathcal{S}) - \epsilon\|. \tag{3}$$

Here, we omit the weighting terms of the loss function for simplicity. The latent $x_0$ can be reconstructed based on the predicted noises.

Let's analyze the inherent information bottleneck at each time step $t$ in the reverse diffusion process. In the reverse diffusion process, the reverse conditional distribution is tractable as [15]:

$$q(x_{t-1}|x_t, x_0) = \mathcal{N}(x_{t-1}|\tilde{\mu}_t, \tilde{\beta}_t\mathbf{I}),$$
$$\text{where} \quad \tilde{\mu}_t = \frac{\sqrt{\bar{\alpha}_{t-1}}\beta_t}{1-\bar{\alpha}_t}x_0 + \frac{\sqrt{\alpha_t}(1-\bar{\alpha}_{t-1})}{1-\bar{\alpha}_t}x_t, \quad \tilde{\beta}_t = \frac{1-\bar{\alpha}_{t-1}}{1-\bar{\alpha}_t}\beta_t. \tag{4}$$

We formulate the Kullback-Leibler (KL) divergence $C_t$ between $q(x_{t-1}|x_t, x_0)$ and the Gaussian prior distribution $p(x_{t-1}) = \mathcal{N}(0, \mathbf{I})$ at step $t-1$ as:

$$C_t = D_{KL}(q(x_{t-1}|x_t, x_0)\|p(x_{t-1})) = \frac{n}{2}(-\log\tilde{\beta}_t - 1 - \tilde{\beta}_t + \tilde{\mu}_t^T\tilde{\mu}_t/n), \tag{5}$$

where $n$ denotes the number of dimension of signal $x$ (*i.e.*, $x_0$, $x_t$, $x_{t-1}$).

In Figure 2 (b), we present the curves that illustrate the KL divergence $C_t$ under different variance ($\beta$) schedules, including linear, sqrt linear, cosine [25, 15], and sqrt schedules [6, 30]. We can see that as the time step $t$ decreases, the KL divergence $C_t$ increases, indicating the information carried by $x_{t-1}$ increases and resulting in increasingly looser information bottlenecks over $x_{t-1}$. According to [2, 14], such time-varying information bottlenecks may play an important role in promoting disentanglement. Different variance ($\beta$) schedules results in different KL divergence curves. We found that these different variance schedules lead to different disentanglement performance (see Subsection 4.4).

Actually, optimizing the loss of conditional diffusion model as in (3) is equivalent to push the reverse conditional distribution $p_\theta(x_{t-1}|x_t, \mathcal{S})$ to approach $q(x_{t-1}|x_t, x_0)$ (see the explanation in Appendix B). By this means, the information bottlenecks $\tilde{C}_t = D_{KL}(p_\theta(x_{t-1}|x_t, \mathcal{S})\|p(x_{t-1}))$ over $x_{t-1}$ tend to approach $C_t = D_{KL}(q(x_{t-1}|x_t, x_0)\|p(x_{t-1}))$ in training for all the time steps. According to the theorem in Appendix C, the information bottleneck over latent $x_{t-1}$ is transferred to the condition $\mathcal{S}$ (*i.e.*, concept tokens). Intuitively, this is because $x_{t-1}$ is controlled by the concept tokens and the network parameters $\theta$, as indicated by $p_\theta(x_{t-1}|x_t, \mathcal{S})$. The concept token representations $\mathcal{S}$ are learnable, and the information bottleneck is transferred to and imposed on $\mathcal{S}$. With time-varying

information bottlenecks, the diffusion process encourages a range of different information capacities on $\mathcal{S}$ during the diffusion process, promoting the disentanglement of concept tokens.

**Discussion** The information bottleneck described above shares a certain similarity with the optimization objective in AnnealVAE [2]. The minimizing objective of AnnealVAE is expressed as follows:

$$\mathcal{L}(\phi, \varphi) = - E_{q_\phi(z|x)}[\log p_\varphi(x|z)] + \gamma \|D_{KL}(q_\phi(z|x)\|p(z)) - C\|, \tag{6}$$

where $\phi, \varphi$ are the parameters of the encoder and decoder; the latent representation and data are denoted as $z, x$, respectively. $C$ is a handcrafted constant used to control the information bottleneck of the latent space. Different factors to be disentangled may contain different amounts of information. Instead of using a constant during training, AnnealVAE dynamically allocates larger amount of information (larger $C$) to the latent units as the training iteration increases, where different factors can be learned at various training stages.

In diffusion models, where the KL divergence characterizes the amount of information, we observe that the amount of information varies in reverse diffusion steps, see Figure 2 (b).

### 3.2.2 Cross Attention for Interaction

The information bottleneck sheds a light on disentangling, acting as an inductive bias for diffusion. However, using the information bottleneck still only has theoretical feasibility. Its effectiveness also relies on the structure design of the diffusion model. We believe that the cross-attention design in conditional diffusion model is crucial for disentanglement, serving as another effective inductive bias.

Our objective is to train an encoder that obtains a set of concept tokens taking an image as input under the guidance of the diffusion model. We take the output of the encoder as the condition of the U-Net of diffusion model for image generation. We incorporate the concept tokens into diffusion model through cross attention. Intuitively, a spatial position of an image is related to several concepts, *e.g.*, object color and shape in Shapes3D. Each spatial feature is composed of several related concept-based representations. Interestingly, cross attention in the U-Net play a similar role, where each spatial feature servers as the query, and the learned concept tokens are used as the keys and values to refine the query. In Subsection 4.4, we validate the necessity of the two inductive biases leading to disentanglement.

## 4 Experiments

### 4.1 Experimental Setup

**Implementation Details** The trainable parts are the encoder and diffusion model. We employ the popular diffusion structure of latent diffusion [30] by default. Without loss of generality, following [30], we use the VQ-reg to avoid arbitrarily high-variance latent spaces and sample images in 200 steps. We adopt the cosine schedule as the variance ($\beta$) schedule in the diffusion model by default. By default, we use a CNN encoder for the image encoder to obtain a set of disentangled concept tokens. We use a CNN encoder similar to that used in [40]. We denote our scheme as EncDiff.

**Training Details** During the training phase of EncDiff, we maintain a consistent batch size of 64 across all datasets. The learning rate is consistently set to $1 \times 10^{-4}$. We adopt the standard practice of employing an Exponential Moving Average (EMA) with a decay factor of 0.9999 for all model parameters. The training hyper-parameters follows DisDiff [40] and DisCo [28]. For each concept token, we follow DisDiff [40] to use a 32 dimensional representation vector. We train EncDiff on a single Tesla V100 16G GPU. A model takes about 1 day for training.

**Datasets** To evaluate the disentanglement performance, we utilize the commonly used benchmark datasets: Shapes3D [19], MPI3D [10] and Cars3D [27]. Shapes3D [19] consists of a collection of 3D shapes. MPI3D is a dataset of 3D objects created in a controlled setting. Cars3D is a dataset consisting of 3D-rendered cars. For real-world data, we conduct our experiments using CelebA, a dataset of celebrity faces with attributes. Our experiments are carried out at a 64×64 image resolution, consistent with previous studies [19, 4, 28, 40].

**Baselines & Metrics** We compare the performance of our method with VAE-based, GAN-based, and diffusion-based methods, following the experimental protocol as in DisCo [28]. The VAE-based

Table 1: Comparisons of disentanglement on the FactorVAE score and DCI disentanglement metrics (mean ± std, higher is better). EncDiff outperforms the state-of-the-art methods with a large margin except on Cars3D.

| Method | Cars3D | | Shapes3D | | MPI3D | |
|---|---|---|---|---|---|---|
| | FactorVAE score↑ | DCI↑ | FactorVAE score↑ | DCI↑ | FactorVAE score↑ | DCI↑ |
| | *VAE-based:* | | | | | |
| FactorVAE [19] | 0.906 ± 0.052 | 0.161 ± 0.019 | 0.840 ± 0.066 | 0.611 ± 0.082 | 0.152 ± 0.025 | 0.240 ± 0.051 |
| $\beta$-TCVAE [4] | 0.855 ± 0.082 | 0.140 ± 0.019 | 0.873 ± 0.074 | 0.613 ± 0.114 | 0.179 ± 0.017 | 0.237 ± 0.056 |
| | *GAN-based:* | | | | | |
| InfoGAN-CR [23] | 0.411 ± 0.013 | 0.020 ± 0.011 | 0.587 ± 0.058 | 0.478 ± 0.055 | 0.439 ± 0.061 | 0.241 ± 0.075 |
| | *Pre-trained GAN-based:* | | | | | |
| LD [32] | 0.852 ± 0.039 | 0.216 ± 0.072 | 0.805 ± 0.064 | 0.380 ± 0.062 | 0.391 ± 0.039 | 0.196 ± 0.038 |
| GS [11] | 0.932 ± 0.018 | 0.209 ± 0.031 | 0.788 ± 0.091 | 0.284 ± 0.034 | 0.465 ± 0.036 | 0.229 ± 0.042 |
| DisCo [28] | 0.855 ± 0.074 | 0.271 ± 0.037 | 0.877 ± 0.031 | 0.708 ± 0.048 | 0.371 ± 0.030 | 0.292 ± 0.024 |
| | *Diffusion-based:* | | | | | |
| DisDiff [40] | **0.976** ± 0.018 | 0.232 ± 0.019 | 0.902 ± 0.043 | 0.723 ± 0.013 | 0.617 ± 0.070 | 0.337 ± 0.057 |
| EncDiff (Ours) | 0.773 ± 0.060 | **0.279** ± 0.022 | **0.999** ± 0.000 | **0.969** ± 0.030 | **0.872** ± 0.049 | **0.685** ± 0.044 |

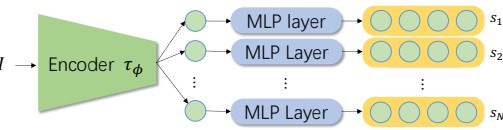

Figure 3: Illustration of the encoder $\tau_\phi$, which transforms an image into a feature vector of dimension $N$, with each dimension (scalar) encoding a disentangled factor. We then use non-shared three-layer MLP layers to map each scalar into a vector (concept token). The concept tokens will be treated as the conditional input to the latent diffusion model with cross attention.

Table 2: Comparisons of disentanglement performance and generation quality in terms of TAD and FID metrics (mean ± std) on real-world dataset CelebA. EncDiff achieves the state-of-the-art performance on both aspects compared to all baselines.

| Model | TAD ↑ | FID ↓ |
|---|---|---|
| $\beta$-VAE [14] | 0.088 ± 0.043 | 99.8 ± 2.4 |
| InfoVAE [43] | 0.000 ± 0.000 | 77.8 ± 1.6 |
| Diff-AE [26] | 0.155 ± 0.010 | 22.7 ± 2.1 |
| InfoDiffusion [34] | 0.299 ± 0.006 | 23.6 ± 1.3 |
| DisDiff [40] | 0.305 ± 0.010 | 18.2 ± 2.1 |
| EncDiff | **0.638 ± 0.008** | **14.8 ± 2.3** |

models we use for comparison are FactorVAE [19] and $\beta$-TCVAE [4], while the GAN-based baselines include InfoGAN-CR [23], GANspace (GS) [11], LatentDiscovery (LD) [32] and DisCo [28]. Each method utilizes scalar-valued representations. DisDiff [40] uses vector-valued representations. EncDiff has two kinds of representations simultaneously. We focus on the scalar-valued in the main paper. We follow DisDiff to set $N$ to 20. For these vector-valued representations, we follow [7, 40, 38] to perform PCA as a post-processing on the representation before evaluation. To assess the potential variability in performance due to random seed selection, we have fifteen runs for each method for reliable evaluation, reporting the mean and variance. Regarding evaluation metrics, we adopt two representative metrics, the FactorVAE score [19] and the DCI [8].

## 4.2 Comparison with the State-of-the-Arts

We compare the disentanglement ability of our EncDiff with the state-of-the-art methods. Table 1 shows quantitative comparison results of disentanglement under different metrics. We can see that EncDiff achieves the best performance on all the datasets except Cars3D, showcasing the model's superior disentanglement ability. EncDiff achieves superior performance by leveraging the strong inductive bias from the diffusion model with cross-attention without using any additional regularization losses. EncDiff also outperforms InfoDiffusion [34] and DisDiff [40] by a significant marginal, even though DisDiff uses complex disentanglement loss and inference the decoder multiple times for prediction sub-gradient fields. On the Cars3D dataset, the quantitative evaluation is not so reliable because some factors, such as color and shape, are not included in the labels. From the visualization in Figure 6 of Appendix E, we can see that EncDiff achieves superior disentanglement compared to DisDiff despite the lower FactorVAE score.

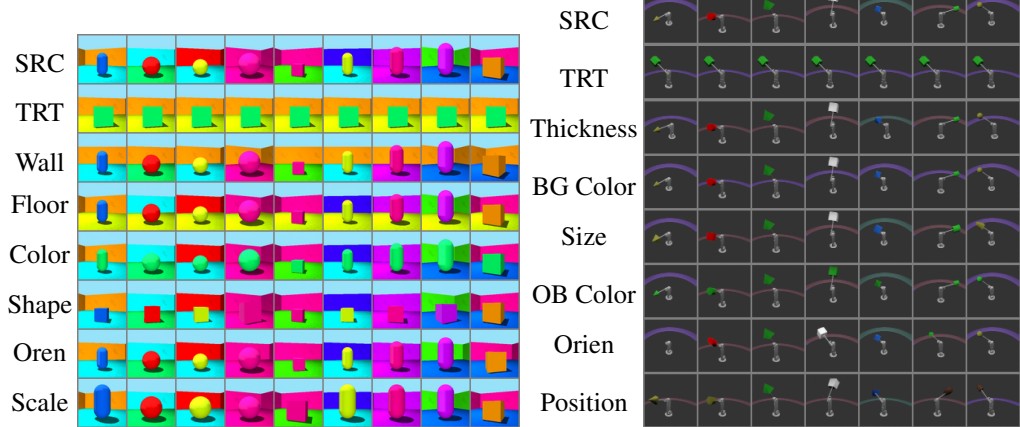

Figure 4: The qualitative results on Shapes3D and MPI3D. The source (SRC) images provide the representations of the generated image. The target (TRT) image provides the representation for swapping. Other images are generated by swapping the representation of the corresponding factor. For Shapes3D, the learned factors on Shapes3D are wall color (Wall), floor color (Floor), object color (Color), and object shape (Shape), orientation (Orien), scale. See Appendix E for more visualizations.

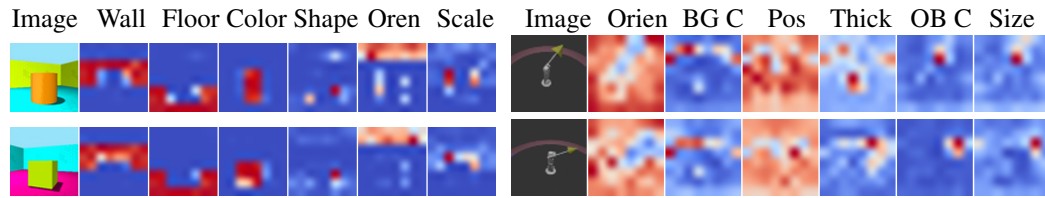

Figure 5: Visualization of the cross-attention maps on Shapes3D and MPI3D. The first column shows the original image while the other columns show the attention masks for different concept tokens. See Appendix F for more visualizations. "Pos" represents "Position".

Moreover, we have conducted experiments on real-world dataset CelebA. Table 2 shows the comparisons of disentanglement performance and generation quality in terms of TAD and FID metrics (mean ± std). EncDiff achieves the state-of-the-art performance on both aspects compared to all baselines.

In addition, our EncDiff achieves superior reconstruction quality (see Appendix G for more details).

### 4.3 Visualization

**Visualization Analysis on the Disentanglement** We qualitatively examine the disentanglement properties of our proposed method. We interchange the concept tokens (factors) of the learned representation of two distinct images and observe the generated images conditioned on these exchanged representations. For illustration purposes, we focus on the widely used Shapes3D dataset from the disentanglement literature and show the results in Figure 4. We can see that our EncDiff successfully isolates factors. Notably, in comparison to VAE-based methods, EncDiff delivers superior image quality quantitatively (see Appendix H).

**Visualization of Learned Cross-Attention Maps** As mentioned in Section 1, our model draws inspiration from the alignment between 'word' tokens (disentangled representations) and spatial features. The alignment is demonstrated by the learned cross-attention maps. We verify whether our learned concept tokens present the disentangled characteristics by visualizing the alignment of concept tokens with spatial positions through cross-attention maps. As depicted in Figure 5, the results of our model exhibit exemplary alignment between concept tokens and spatial positions. Different concept tokens are associated with varying attended spatial regions, corresponding to different semantics that are comprehensible by humans, such as the region of "Floor" and "Color" for the images from Shapes3D.

Table 3: Influence of the two inductive biases. For *EncDec w/o Diff*, we replace the diffusion model with a decoder while cross-attention is preserved. For *EncDiff w/ AdaGN*, we replace the cross-attention with AdaGN.

| Method | FactorVAE score↑ | DCI↑ |
|---|---|---|
| EncDec w/o Diff | $0.537 \pm 0.074$ | $0.178 \pm 0.050$ |
| EncDiff w/ AdaGN | $0.911 \pm 0.101$ | $0.637 \pm 0.068$ |
| EncDiff | $\mathbf{0.999} \pm 0.000$ | $\mathbf{0.969} \pm 0.030$ |

Table 4: Ablation study on the influence of the variance ($\beta$) schedule. We use four kinds of variance schedules: sqrt, cosine, linear, and sqrt linear.

| Method | FactorVAE score↑ | DCI↑ |
|---|---|---|
| EncDiff w/sqrt | $0.997 \pm 0.011$ | $0.950 \pm 0.041$ |
| EncDiff w/sqrt linear | $0.988 \pm 0.026$ | $0.924 \pm 0.050$ |
| EncDiff w/linear | $0.999 \pm 0.002$ | $0.930 \pm 0.045$ |
| EncDiff w/cosine | $\mathbf{0.999} \pm 0.001$ | $\mathbf{0.969} \pm 0.030$ |

## 4.4 Ablation Study

In our framework, in order to analyze and understand the key factors contributing to the advancement of disentangled representation learning, we conduct ablation studies covering three aspects in the design: 1) whether to use diffusion as the decoder; 2) whether to use cross attention as the bridge for interaction; 3) influence of different variance ($\beta$) schedules. We conduct these ablation studies on the Shapes3D dataset. Please see Appendix H for more ablation studies.

**Using Diffusion as Decoder or Not** To validate whether the use of a diffusion model as an inductive bias for disentangled representation learning is crucial, we employ a network structure similar to the U-Net in our used diffusion model as the decoder, utilizing reconstruction $l_2$ loss to optimize the entire network. We designed a variant (EncDec w/o Diff) of EncDiff to have an autoencoder-like structure, by reusing the image encoder as encoder and the lower half of the U-Net structure as decoder for reconstruction. In contrast to EncDiff, we discard the multiple step diffusion process and only run once feedforward inference to get the reconstruction. If the autoencoder's performance drops significantly, this will provide evidence of the importance of the diffusion process instead of the U-Net architecture. Specifically, we remove the encoder part of the U-Net and the skip connection between it and the decoder part of the U-Net. We then feed the U-Net decoder with a randomly initialized learnable spatial tensor to maintain the structure of the decoder U-Net. Similarly to EncDiff, the encoded disentangled representations are input to the decoder through cross-attention (CA). We refer to this scheme as *EncDec w/o Diff*. Table 3 shows the results. The performance of *EncDiff* with diffusion significantly outperforms *EncDec w/o Diff* by 0.46 and 0.79 in terms of FactorVAE score and DCI, respectively. This indicates that inductive bias from diffusion modelling is crucial for achieving effective disentanglement.

**Using Cross Attention for Interaction** To incorporate the image representation into the diffusion model as a condition, we use cross attention by treating each disentangled representation as a conditional token (similar to the use of 'word' token in text-to-image generation in stable diffusion model [30]). As an alternative, similar to that in Diff-AE [26] and InfoDiffusion [34], we use adaptive group normalization (AdaGN) to incorporate the representation vector (by concatenating the concept tokens) to modulate the spatial features. We name this scheme as *EncDiff w/ AdaGN*. Table 3 presents the results. We can see that *EncDiff w/ AdaGN* is inferior to *EncDiff*, with a significant decrease of 0.33 in terms of DCI. Cross-attention facilitates the alignment of each concept token with the corresponding spatial features, akin to the alignment of the 'word' token to spatial features in the text-to-image generation. In contrast, AdaGN did not efficiently promote disentanglement.

**Influence of Different Variance ($\beta$) Schedules** We investigate the influence of the different variance ($\beta$) schedules, including including linear, sqrt linear, cosine [25, 15], sqrt schedules [6, 30], on the disentanglement performance. From Table 4, we can see that distinct schedules result in different performance, demonstrating the influence on disentanglement of different information bottleneck schedules. Note that the FactorVAE scores are all very high and cannot well reflect the performance. We prefer to use DCI metric here for evaluation. We can see that the cosine schedule performs the best and we adopt it by default. The linear schedule approaches that of the sqrt linear in terms of the curve shape (see Figure 2 (b)) and they achieve the similar performance in terms of DCI.

**Scalar-valued vs. Vector-valued Manners** We treat each dimension of the encoded feature vector as a disentangled factor, followed by a mapping to a concept token (vector) for each factor. Another design alternative is to split the feature vector into $N$ chunks, with each chunk being a concept token, similar to DisDiff [40]. We name this vector-valued design and refer to it by DisDiff-V. Table 5 shows that EncDiff outperforms EncDiff-V. The intermediate scalar design in EncDiff may serve a

Table 5: Ablation study on the two design alternatives on obtaining the token representations.

| Method | FactorVAE score ↑ | DCI↑ |
|---|---|---|
| EncDiff-V | $0.999 \pm 0.000$ | $0.900 \pm 0.045$ |
| EncDiff | $\mathbf{0.999} \pm 0.001$ | $\mathbf{0.969} \pm 0.030$ |

Table 6: Ablation study on the space applying diffusion model.

| Method | FactorVAE score↑ | DCI↑ |
|---|---|---|
| EncDiff pixel | $1.0 \pm 0.000$ | $0.981 \pm 0.015$ |
| EncDiff | $0.999 \pm 0.000$ | $0.969 \pm 0.030$ |

Table 7: Computational complexity comparison.

| Method | Params.↓ (M) | FLOPs↓ (M) | Time↓ (s) |
|---|---|---|---|
| Diff-AE [26] | 67.8 | 3955.1 | 31.0 |
| DisDiff [40] | 57.1 | 5815.8 | 35.3 |
| EncDiff | 42.3 | 2898.5 | 11.8 |

bottleneck role and contribute to the disentanglement. We think that the vector-based representation potentially extracts more information and hence enforces a looser bottleneck than scalar-valued representation. Note that the more information encoded, there is a higher probability that the encoded information is correlated, which is contradictory to disentanglement. Therefore, the tighter bottleneck from scalar-valued representation leads to (a slightly) better performance.

**Results on Pixel Space** As stated in Section 3.2.1, our analysis is applicable in pixel space as well. In order to verify this, we trained EncDiff directly in pixel space on the Shapes3D dataset, which we denote the scheme as EncDiff pixel. The results of the disentanglement analysis are presented in Table 6. The performance of our framework in pixel space remains robust, indicating that the operation in latent space is not a critical factor for achieving effective disentanglement.

### 4.5 Computational Complexity

We compare the computational complexity of Diff-AE [26], DisDiff [40], and our EncDiff in terms of the parameters (Params.), floating-point operations (FLOPs), and inference time (seconds/sample) for sampling an image. As shown in Table 7, our EncDiff demonstrates much higher computational efficiency than Diff-AE and DisDiff.

## 5 Limitations

Our method operates in a fully unsupervised manner and exhibits strong disentanglement capability on simple datasets. Similar to other disentanglement-based methods [14, 19, 5, 23, 37], obtaining satisfactory performance on complex data remains a challenge. As a diffusion-based method, the generation speed of EncDiff is faster than DisDiff [40]. However, it is still slower when compared to VAE-based and GAN-based methods. More effective sampling strategies, as employed in DPM-based methods, could be utilized for accelerating in the future.

## 6 Conclusion

This paper unveils a fresh viewpoint, demonstrating that diffusion models with cross-attention can serve as a strong inductive bias to foster disentangled representation learning. Within our framework EncDiff, we reveal that the diffusion model structure with cross-attention can drive an image encoder to learn superior disentangled representations, even without any regularization. Our comprehensive ablation studies demonstrate that the strong capability is mainly attributed to diffusion modelling and cross-attention interaction. This work will inspire further investigations in diffusion for disentanglement, paving the way for sophisticated data analysis, understanding, and generation.

## 7 Acknowledgement

We thank all the anonymous reviewers for their constructive and helpful comments, which have significantly improved the quality of the paper. The work was partly supported by the National Natural Science Foundation of China (Grant No. 62088102).

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

## A    Impact Statements

This paper presents work whose goal is to advance the field of disentanglement learning. Our research is designed to be a positive force for innovation purpose. Viewed from a societal lens, the potential negative impacts is the malicious use of the models. This highlights the critical necessity of incorporating ethical considerations in the utilization of our method for responsible AI.

## B    Optimization of the Reverse Conditional Probability $p_\theta(x_{t-1}|x_t, \mathcal{S})$

Two distinct approaches exist for decomposing the loss function, Variational Lower Bound(VLB), of the diffusion model. The derivations presented herein closely follow the methodology outlined by Ho et al. [15]. Without loss of generality, we could incorporate the conditional input denoted by $\mathcal{S}$ to the diffusion.

For the first decomposition alternative, we can derive the following equations:

$$
\begin{aligned}
\mathcal{L} &= \mathbb{E}_q\left[-\log\frac{p_\theta(x_{0:T}|\mathcal{S})}{q(x_{1:T}|x_0)}\right] \\
&= \mathbb{E}_q\left[\log\frac{\Pi_{t=1}^T q(x_t|x_{t-1})}{p(x_T)\Pi_{t=1}^T p_\theta(x_{t-1}|x_t,\mathcal{S})}\right] \\
&= \mathbb{E}_q\left[-\log p(x_T) + \Sigma_{t\geq 1}\log\frac{q(x_t|x_{t-1})}{p_\theta(x_{t-1}|x_t,\mathcal{S})}\right] \\
&= \mathbb{E}_q\left[-\log p(x_T) - \Sigma_{t>1}\log\frac{p_\theta(x_{t-1}|x_t,\mathcal{S})}{q(x_t|x_{t-1})} - \log\frac{p_\theta(x_0|x_1)}{q(x_1|x_0)}\right].
\end{aligned}
\tag{7}
$$

By applying Bayes' Rule and the Markov property of the diffusion process, we conclude that

$$
q(x_{t-1}|x_t,x_0) = \frac{q(x_t|x_{t-1},x_0)q(x_{t-1}|x_0)}{q(x_t|x_0)} = \frac{q(x_t|x_{t-1})q(x_{t-1}|x_0)}{q(x_t|x_0)}.
\tag{8}
$$

This leads to

$$
\begin{aligned}
\mathcal{L} &= \mathbb{E}_q\left[-\log p(x_T) - \Sigma_{t>1}\log\frac{p_\theta(x_{t-1}|x_t,\mathcal{S})}{q(x_{t-1}|x_t,x_0)}\frac{q(x_{t-1}|x_0)}{q(x_t|x_0)} - \log\frac{p_\theta(x_0|x_1)}{q(x_1|x_0)}\right] \\
&= \mathbb{E}_q\left[-\log\frac{p(x_T)}{q(x_T|x_0)} - \Sigma_{t>1}\log\frac{p_\theta(x_{t-1}|x_t,\mathcal{S})}{q(x_{t-1}|x_t,x_0)} - \log p_\theta(x_0|x_1)\right] \\
&= \mathbb{E}_q\left[D_{KL}(q(x_T|x_0)||p(x_T)) + \Sigma_{t>1}D_{KL}(q(x_{t-1}|x_t,x_0)||p_\theta(x_{t-1}|x_t,\mathcal{S})) - \log p_\theta(x_0|x_1)\right].
\end{aligned}
\tag{9}
$$

For the second alternative, the loss function is derived to be $\mathbb{E}_{x_0,\epsilon,t}\|\epsilon_\theta(x_t,t,\mathcal{S}) - \epsilon\|$ [15].

The optimization of conditional diffusion model by predicting the noises through minimizing $\mathbb{E}_{x_0,\epsilon,t}\|\epsilon_\theta(x_t,t,\mathcal{S}) - \epsilon\|$ is thus equivalent to minimizing the second term of (9), *i.e.*, $\Sigma_{t>1}D_{KL}(q(x_{t-1}|x_t,x_0)||p_\theta(x_{t-1}|x_t,\mathcal{S}))$, which pushes the reverse conditional probability $p_\theta(x_{t-1}|x_t,\mathcal{S})$ to approach $q(x_{t-1}|x_t,x_0)$.

## C    The Transfer of Information Bottleneck

**Theorem C.1.** *The Kullback-Leibler divergence is invariant under a differentiable mapping $f$, i.e .*

$$
D_{KL}(p(x)|q(x)) = D_{KL}(p(S)|q(S))
$$

*where $x = f(S)$ is a differentiable function between $x$ and $S$ and $p(x)$ and $q(x)$ are the probability density functions of the probability distributions $P$ and $Q$, respectively.*

*Proof*: The Kullback-Leibler divergence (KL divergence) is defined as

$$
D_{KL}(p(x)|q(x)) = \int p(x)\log\frac{p(x)}{q(x)}dx
$$

According to the change of variable theorem, we have

$$
\begin{aligned}
p(x) &= p(f(x))|f'(x)| = p(S)|f'(x)| \\
q(x) &= q(f(x))|f'(x)| = q(S)|f'(x)|
\end{aligned}
\tag{10}
$$

where $|f'(x)|$ denotes the Jacobian of $f(x)$, $p(S)$ is the corresponding distribution of $p(x)$ under the mapping $f$. $q(S)$ is the corresponding distribution of $q(x)$ under the mapping $f$. Combine the two equations, we have

$$
D_{KL}(p(x)|q(x)) = \int p(f(x))|f'(x)| \log \frac{p(f(x))|f'(x)|}{q(f(x))|f'(x)|} dx = \int p(f(x)) \log \frac{p(f(x))}{q(f(x))} |f'(x)| dx.
\tag{11}
$$

Considering the property of integral that for $S = f(x)$ we have $dS = |f'(x)|dx$. We then have the following:

$$
D_{KL}(p(x)|q(x)) = \int p(S) \log \frac{p(S)}{q(S)} dS = D_{KL}(p(S)|q(S)).
\tag{12}
$$

This indicates that the information bottleneck on $x$ is transferable to the input $S$ of the function.

## D    Implementation Details

For our EncDiff (scalar-valued), following the approach of [22], each dimension of the encoded feature vector is treated as a disentangled factor. Each factor is then mapped to a vector (i.e., concept token) using non-shared MLP layers, as illustrated in Figure 3. For vector-valued EncDiff (EncDiff-V), inspired by DisDiff [40], we partition the feature vector into $N$ (e.g., 20) chunks, referred to as concept tokens, to encode different factors.

**Image Encoder Architecture** To ensure an equitable comparison, we employ the encoder architecture, consistent with DisCo [28] and DisDiff [40]. The encoder specifications are detailed in Table 8.

Table 8: Encoder architecture used in EncDiff.

| |
|---|
| Conv $7 \times 7 \times 3 \times 64$, stride $= 1$ |
| ReLu |
| Conv $4 \times 4 \times 64 \times 128$, stride $= 2$ |
| ReLu |
| Conv $4 \times 4 \times 128 \times 256$, stride $= 2$ |
| ReLu |
| Conv $4 \times 4 \times 256 \times 256$, stride $= 2$ |
| ReLu |
| Conv $4 \times 4 \times 256 \times 256$, stride $= 2$ |
| ReLu |
| FC $4096 \times 256$ |
| ReLu |
| FC $256 \times 256$ |
| ReLu |
| FC $256 \times K$ |

**Diffusion U-Net Architecture** The diffusion architecture adheres to the design principles of latent diffusion [30] and DisDiff [42]. Table 9 provides a detailed overview of the network structure, similar to the structure of Latent Diffusion Probabilistic Model.

We pretrain VQ-VAE in diffusion. Then the diffusion network and our image encoder are jointly trained.

## E    More Visualizations

We present qualitative results for Cars3D in Figure 6. Notably, on the synthetic dataset Cars3D, EncDiff demonstrates the acquisition of disentangled representations, presenting better disentanglement ability than DisDiff [40]). We can see that our EncDiff can capture the factors of "Color", "Azimuth",

Table 9: U-Net architecture used in EncDiff.

| Parameters | Shapes3D / Cars3D/ MPI3D /CelebA |
|---|---|
| Base channels | 16 |
| Channel multipliers | [ 1, 2, 4, 4 ] |
| Attention resolutions | [ 1, 2, 4] |
| Attention heads num | 8 |
| Model channels | 64 |
| Dropout | 0.1 |
| Images trained | 0.48M / 0.28M / 1.03M |
| $\beta$ scheduler | Cosine (Sqrt/Sqrt linear/Linear) |
| Training T | 1000 |
| Diffusion loss | MSE with noise prediction $\epsilon$ |

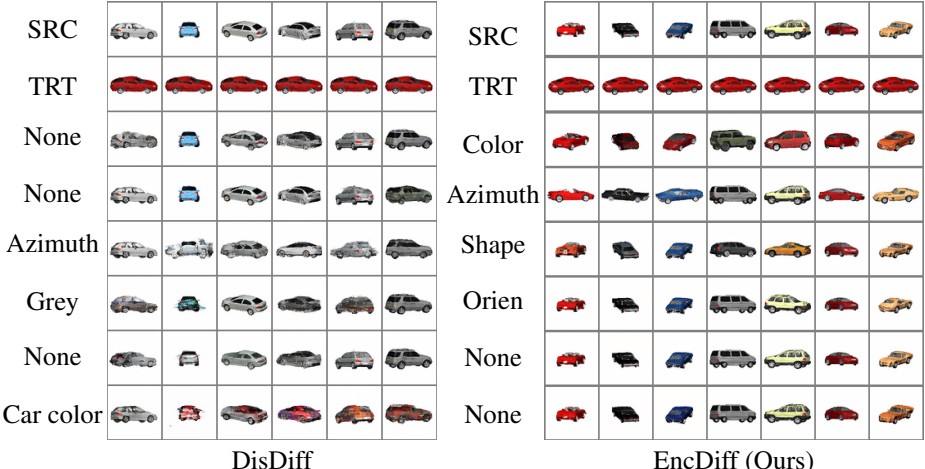

Figure 6: The qualitative comparison on Cars3D. The source (SRC) images provide the representations of the generated image. The target (TRT) image provides the representation for swapping. Other images are generated by swapping the representation of the corresponding factor. "Orien" refers to Orientation. We can see that our EncDiff can capture the factors of "Color", "Azimuth", and "Shape" while DisDiff failed to capturing them.

and "Shape" while DisDiff failed to capturing them. Furthermore, we include three rows of images demonstrating the manipulation of representations lacking informative content, denoted as "None".

We present the qualitative outcomes for MPI3D in Figure 7. Remarkably, on the challenging disentanglement dataset MPI3D, EncDiff showcases its ability to obtain disentangled representations.

# F    More Visualizations on Attention Maps

We showcase the visualization of attention maps for our model's disentangled representations on the Cars3D dataset, as depicted in Figure 8. Similar to EncDiff, the attention maps provide insights into the acquisition of disentangled representations in the synthetic setting of Cars3D. Notably, our attention maps reveal distinct alignments between concept tokens and spatial features, demonstrating the disentangled characteristics learned by our model.

For the MPI3D dataset, Figure 8 demonstrate the qualitative outcomes of attention map visualization. In this challenging disentanglement scenario, EncDiff excels in acquiring disentangled representations. The attention maps further illustrate the alignment between concept tokens and spatial positions, affirming the model's ability to disentangle complex features in MPI3D.

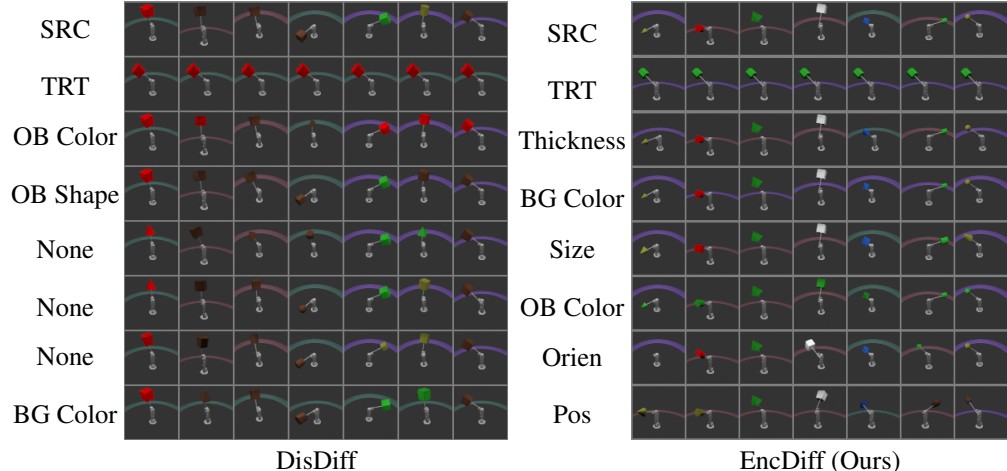

DisDiff                           EncDiff (Ours)

Figure 7: The qualitative results on MPI3D. The source (SRC) images provide the representations of the generated image. The target (TRT) image provides the representation for swapping. Other images are generated by swapping the representation of the corresponding factor. The learned factors on MPI3D are thickness, BG (Background) color, object (OB) color, and object (OB) shape, orientation (Orien), Pos (Background bar position).

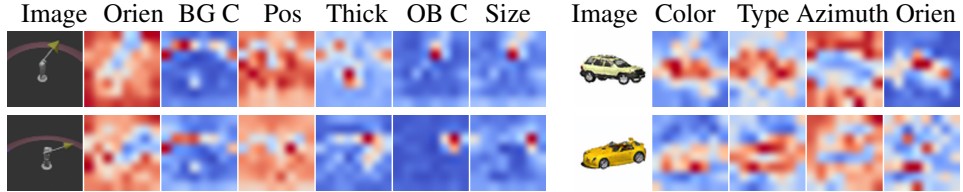

Figure 8: Visualization of the cross-attention maps on MPI3D and Cars3D. The first column shows the original image, while the other columns show the attention masks for different concept tokens.

## G   Autoencoding Reconstruction Quality

**Autoencoding Reconstruction Quality** To investigate the autoencoding reconstruction quality of EncDiff, we conduct the same quantitative experiments with PDAE, DisDiff and Diff-AE. We follow them to evaluate the reconstruction quality using averaged SSIM, LPIPS, and MSE. The results are shown in Table 10. It is evident that EncDiff outperforms DisDiff on all metrics. EncDiff achieves the state-of-the-art performance of SSIM and LPIPS with a strong disentanglement capability.

Table 10: Reconstruction quality comparison on the Shape3D dataset.

| Method | SSIM↑ | LPIPS↓ | MSE↓ | DCI↑ | Factor VAE↑ |
|---|---|---|---|---|---|
| PDAE | 0.9830 | 0.0033 | 0.0005 | 0.3702 | 0.6514 |
| Diff-AE | 0.9898 | 0.0015 | $\mathbf{8.1983e-05}$ | 0.0653 | 0.1744 |
| DisDiff | 0.9484 | 0.0006 | $9.9293e-05$ | 0.723 | 0.902 |
| EncDiff (Ours) | **0.9997** | **0.0003** | $9.6299e-05$ | **0.999** | **0.969** |

## H   More Ablation Related with EncDiff

**Measuring on Scalar-valued vs. Vector-valued Representations in our EncDiff** For EncDiff, each dimension of the scalar-valued representation is mapped to a representation vector, resulting in two representations in EncDiff: a scalar-valued one and a mapped vector-valued one. From Table 11, we can see that these two representations have similar performance.

Table 11: Comparison of the two different representations (scalar-valued vs vector-valued) in EncDiff.

| Method | FactorVAE score ↑ | DCI↑ |
|---|---|---|
| EncDiff (Vector) | $0.998 \pm 0.006$ | $0.955 \pm 0.030$ |
| EncDiff (Scalar) | $\mathbf{0.999} \pm 0.001$ | $\mathbf{0.969} \pm 0.030$ |

**Influence of the Number of Concept Tokens** Similar to other disentanglement methods, the number of disentangled latent units influences performance. To study such an effect on EncDiff, we train our model with the following number of tokens: 5, 10, 15, 20 and 30, respectively. As shown in Table 12, when the number of tokens is fewer than the number of ground truth factors, the performance significantly drops. With the use of more tokens, the performance improves. In accordance with the setup proposed by [39], EncDiff adopts the default setting of 20 tokens for a fair comparison with other methods [39, 40, 28].

Table 12: Influence of the number of concept tokens in EncDiff.

| # Tokens | FactorVAE score ↑ | DCI ↑ |
|---|---|---|
| 5 | $0.605 \pm 0.084$ | $0.536 \pm 0.073$ |
| 10 | $0.985 \pm 0.032$ | $0.900 \pm 0.058$ |
| 15 | $0.996 \pm 0.015$ | $0.936 \pm 0.039$ |
| 20 | $\mathbf{0.999} \pm 0.001$ | $\mathbf{0.969} \pm 0.030$ |
| 30 | $0.995 \pm 0.014$ | $0.961 \pm 0.039$ |

**Efficacy of Additional Regularization** We are wondering whether additional regularization can further promote the disentanglement. To validate this, we conducted the following experiments to investigate the effectiveness of incorporating two types of constraints: sparsity and orthogonality, respectively.

We explored the orthogonality constraint proposed in [3], which enforces orthogonality from a group theory perspective. We adapted their method of transforming representations using Euler encoding to enforce orthogonality within EncDiff. We adopted the official implementation on github to modify EncDiff, denoted as EncDiff with [3]. We integrated the sparsity regularization terms proposed in [9] into EncDiff to facilitate disentanglement, denoted as EncDiff with [9]. We take the techniques proposed in [41], which involve matrix decomposition and matrix exponentiation to construct orthogonal matrices. We replaced the scalar MLP mappings with a series of learnable orthogonal vectors to ensure orthogonality in the representations, denoted as EncDiff with [41].The results are shown in Table 13. We observed that the regularization can slightly improve the performance further on our EncDiff. For simplicity, we do not incorporate any regularization on all other results.

Table 13: Ablation study on the additional regularization over EncDiff. We use a CNN encoder by default.

| Method | FactorVAE score↑ | DCI↑ |
|---|---|---|
| EncDiff w/[3] | $0.999 \pm 0.000$ | $0.965 \pm 0.040$ |
| EncDiff w/[41] | $0.999 \pm 0.002$ | $0.972 \pm 0.029$ |
| EncDiff w/[9] | $0.999 \pm 0.001$ | $\mathbf{0.975} \pm 0.031$ |
| EncDiff | $\mathbf{0.999} \pm 0.001$ | $0.969 \pm 0.030$ |

**Influence of the Image Encoder Architecture** To investigate the influence of encoder design, we use a powerful encoder to replace the CNN encoder used in the EncDiff. We adopt a transformer encoder with a set of learnable tokens as the disentangled representations, as introduced in [39], which are refined through cross-attention. We refer to this scheme *EncDiff w/Trans*. For fair comparison, the model size of the encoders is similar. As shown in Table 14, *EncDiff w/Trans* is comparable to EncDiff. When considering the performance gap between DisDiff [40] and our EncDiff, the influence of encoder structures is small and is not the key factor for influencing disentangling capabilities. A similar phenomenon is observed in vector-valued one.

Table 14: Ablation study on image encoder. EncDiff w/Trans denotes the scheme in which we replace CNN encoder with a transformer encoder.

| Method | FactorVAE score↑ | DCI↑ |
|---|---|---|
| DisDiff [40] | $0.902 \pm 0.043$ | $0.723 \pm 0.013$ |
| EncDec w/Trans | $0.962 \pm 0.034$ | $0.898 \pm 0.033$ |
| EncDiff | $\mathbf{0.999} \pm 0.001$ | $0.969 \pm 0.030$ |

# I  More Ablation Study on EncDiff-V

We also perform ablation study related with EncDiff-V to validate the effects of the two inductive bias, the inefficacy of additional regularization. Similar trends as EncDiff are observed.

**Ablation on Two Inductive Bias of EncDiff-V** In alignment with the main paper, we also conducted an experiment to assess the effectiveness of the two inductive biases in EncDiff-V. We adopt the same decoder used in Section 4.4 to substitute the diffusion in EncDiff-V. We denote this model as EncDec-V w/o Diff. On the other hand, to study the effectiveness of cross-attention, we use the same conditional decoder of EncDiff w/ AdaGN in EncDiff-V, denoted as EncDiff-V w/ AdaGN. The performance of both of these two models drops significantly, as indicated by the results in Table 15.

Table 15: Ablation study on the influence of the two different inductive biases of EncDiff-V. For *EncDec-V w/o Diff*, we replace the diffusion model with a decoder while cross attention is preserved for the interaction. For *EncDiff-V w/ AdaGN*, we replace the cross attention with AdaGN.

| Method | FactorVAE score↑ | DCI↑ |
|---|---|---|
| EncDec-V w/o Diff | $0.682 \pm 0.092$ | $0.246 \pm 0.073$ |
| EncDiff-V w/ AdaGN | $0.956 \pm 0.039$ | $0.520 \pm 0.083$ |
| EncDiff-V | $0.999 \pm 0.000$ | $0.900 \pm 0.045$ |

# J  Ablation Study on MPI3D

Besides Shapes3d, we conducted the main ablation study on another dataset MPI3D. The results are shown in Table 16 and 17 below. We can observe that the trends are consistent with that on Shapes3D (see Table 3).

Table 16: Influence of the two inductive biases on MPI3D. For EncDec w/o Diff, we replace the diffusion model with a decoder while cross-attention is preserved. For EncDiff w/ AdaGN, we replace the cross-attention with AdaGN.

| Method | FactorVAE score↑ | DCI↑ |
|---|---|---|
| EncDiff w/o diff | $0.355 \pm 0.075$ | $0.143 \pm 0.038$ |
| EncDiff w/ AdaGN | $0.592 \pm 0.111$ | $0.268 \pm 0.062$ |
| EncDiff | $0.872 \pm 0.049$ | $0.685 \pm 0.044$ |

Table 17: Ablation study on the two design alternatives on obtaining the token representations.

| Method | FactorVAE score↑ | DCI↑ |
|---|---|---|
| EncDiff-V | $0.863 \pm 0.075$ | $0.629 \pm 0.047$ |
| EncDiff | $0.872 \pm 0.049$ | $0.685 \pm 0.044$ |

# K  Computational Complexity on More Methods

The computational complexity of the VAE-based and GAN-based methods are also listed as shown in Table. The computational complexity and inference time VAEs and GANs still have strength, but diffusion has much better generation and disentangling ability. Among the diffusion models, our EncDiff has better generation and disentangling ability but less computational cost and inference time.

Table 18: Computational complexity comparison.

| Method | Params.↓ (M) | FLOPs↓ (M) | Time↓ (s) |
|---|---|---|---|
| FactorVAE [19] | 11.9 | 892.1 | < 1 |
| BetaTCVAE [4] | 7.9 | 542.1 | < 1 |
| DisCo [28] | 12 | 907.2 | < 1 |
| EncDiff | 42.3 | 2898.5 | 11.8 |

## L    Comparison with Diff-AE

Diff-AE shows disentanglement qualities. One maybe interested in a comparison of disentanglement between Diff-AE and EncDiff. The results of Diff-AE on Shapes3D as shown in Table 19, which is consistent with the trends on CelebA (see Table 2 in the main paper), our method outperforms Diff-AE with a large margin on Shapes3D.

Table 19: Performance comparison with Diff-AE on Shapes3D.

| Method | FactorVAE score↑ | DCI↑ |
|---|---|---|
| Diff-AE | 0.1744 | 0.0653 |
| EncDiff | 0.872 | 0.685 |

## M    EncDiff for Stable Diffusion, i.e., EncDiff (SD)

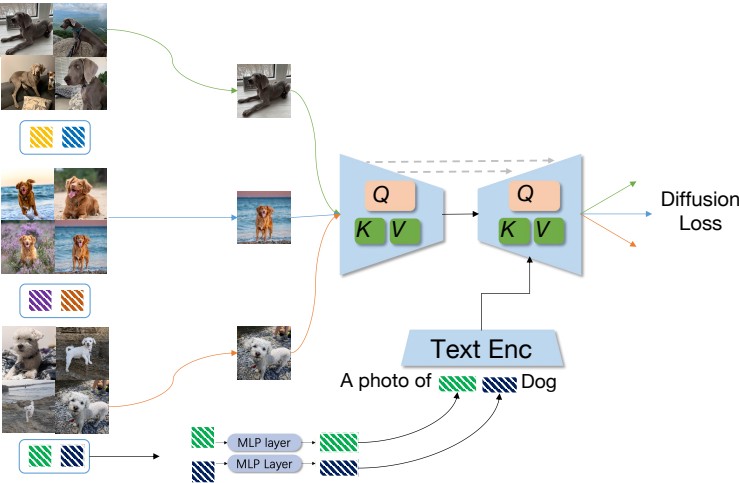

Figure 9: Illustration of applying EncDiff for disentangling DreamBooth i.e., EncDiff (SD).

In order to discuss the disentanglement of EncDiff on real-world data, a newly designed architecture is introduced in Figure 9, named EncDiff(SD).In this architecture, we use a strong pretrained model (stable diffusion v1.4) to replace the original diffusion in EncDiff. Motivated by DreamBooth [31] for customized representation learning, we assign several latent units for each object for inverting and learning object semantic representation. As shown in Figure 9, in order to disentangle these latent units, we use a set of non-shared MLPs to map these latent units into concept tokens. Different from EncDiff, the instance for disentangling is not an image but the semantics of objects. The target is to disentangle concepts or properties (e.g., color, long-hair, big-eared) from the inverted objects (dog).

Figure 10 shows sampling process of EncDiff (SD), we denote the image on the left as image1 and image on the right as image2. We use a prompt "A <token from image1, token from image2> dog is playing a blue ball" to sample image. The <token from image1> encodes the color of the dog. The <token from image2> encodes the type of the dog. We can sample images of new objects that combine color of dog1 and type of dog2.

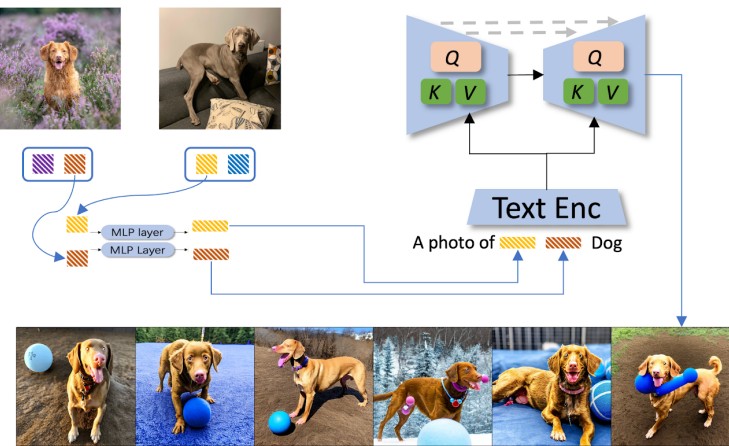

Figure 10: Sampling of EncDiff(SD). We can combine different concepts by sampling EncDiff (SD) with combined tokens from different images.

We take the new prompt as the condition of Stable Diffusion, so that we can use diffusion as an inductive bias for disentangling. The process above is the same as EncDiff. Firstly, to disentangle the objects from the background, we replace the Diffusion Loss with DisenBooth Loss. Secondly, to learn multiple concepts, we follow Custom Diffusion to finetune only the KV layer in cross attention. Lastly, for the ease of learning disentangled concept with a few images, we take an image from each object to construct a training batch. The results are shown in Figure 11, 12, 13. Our model also demonstrates the ability to disentangled semantic factors on real and complex data.

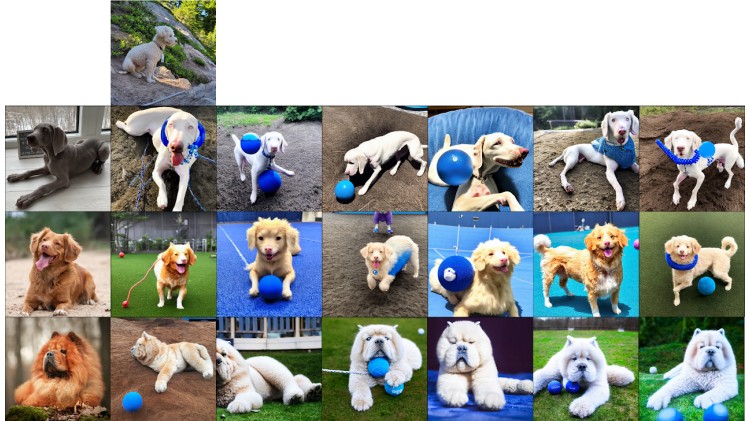

Figure 11: Sampling of EncDiff(SD). Prompt: ”A <new concept tokens> dog is playing a blue ball“. The images of the first column provide source representation and the image (target image) in the first row provides the target representation. The concept write color is disentangled in representation of target image.

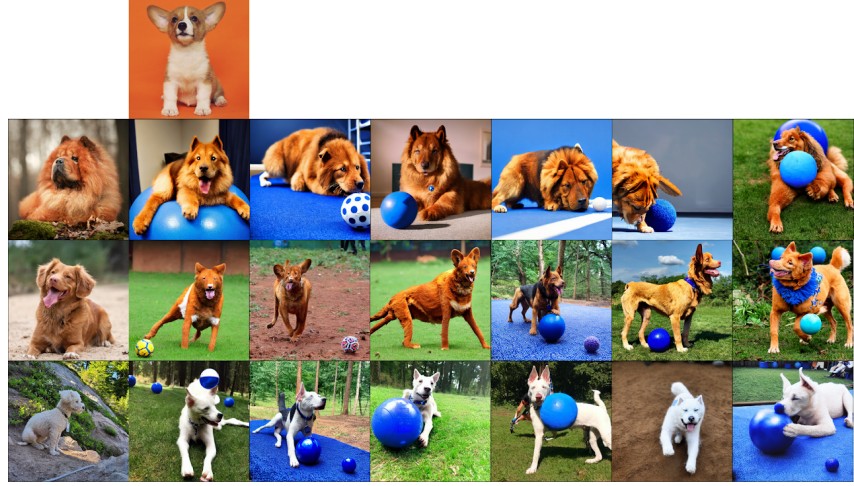

Figure 12: Sampling of EncDiff(SD). We can combine different concept in images by sampling. Prompt "A <new concept tokens> dog is playing a blue ball". The images of the first column provide source representation and the image (target image) in the first row provides the target representation. The concept long ear is disentangled in representation of target image.

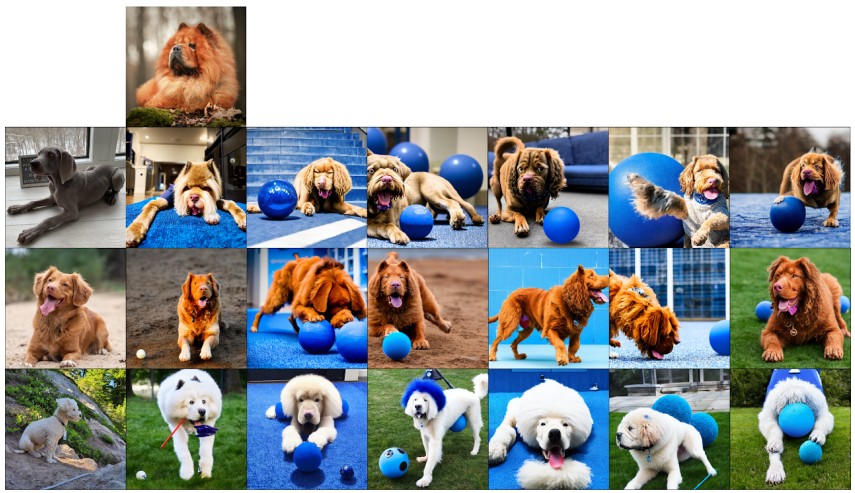

Figure 13: Sampling of EncDiff(SD). We can combine different concept in images by sampling. Prompt "A <new concept tokens> dog is playing a blue ball". The images of the first column provide source representation and the image (target image) in the first row provides the target representation. The concept long hair is disentangled in representation of target image.

