# OpenReview forum: "Diffusion Model with Cross Attention as an Inductive Bias for Disentanglement"
_NeurIPS.cc/2024/Conference — NeurIPS 2024 spotlight_

### Official Review · Reviewer_Mm6Q · 2024-07-05

**Soundness:** 2
**Presentation:** 2
**Contribution:** 2
**Rating:** 6
**Confidence:** 3

**Summary:**

The paper introduces a new approach using diffusion models with cross-attention to improve the learning of disentangled representations in images. By encoding an image into concept tokens and using cross-attention to connect the encoder and the U-Net of the diffusion model, the authors show that the diffusion process creates information bottlenecks that promote disentanglement.

**Strengths:**

- The paper is clear and well-written

- In general, research disentanglement with diffusion models is promising and a good direction for improving the disentanglement community.

- The presented inductive biases are somewhat novel and produce empirically strong disentanglement results with a more straightforward framework.

**Weaknesses:**

- The ablation studies are conducted only on one dataset. I am concerned that the results could be inconsistent between different languages.

- The model does not compare itself to Diff-AE[24] because it does not explicitly disentangle the data. However, to show improvement, it is essential to compare the models to see if they are better since they are similar in many aspects. Diff-AE does show disentanglement qualities in the paper. In addition, a comparison of high-quality Diff-AE and the suggested method could be interesting.

- The resource comparison between the method and other competitive methods is unclear and lacks empirical results.

**Questions:**

- It is unclear to me if section 3.1 is claimed to be the background or contribution of the paper?

- In Eq.4, the author introduces the reverse diffusion process. However, the equation describes the probability of $x_t$ given $x_{t-1}$, which is, as far as I know, the forward process. Is there a typo maybe?

**Limitations:**

No significant limitations compared to current disentanglement methods.

---

> ### Author Rebuttal · Authors · 2024-08-07
>
> Thank you very much for your constructive suggestions, and positive feedback on the paper novelty, strong disentanglement results, and writing. We have carefully considered your valuable suggestions and comments and will incorporate them into our revised manuscript. Please find our detailed responses below.
>
> **Q1**: The ablation studies are conducted only on one dataset. I am concerned that the results could be inconsistent between different languages.
>
> **A1**: Thank you for your suggestion. We also conducted the main ablation study on another dataset MPI3D. The results are shown in Table C below. We can observe that the trends are consistent with that on Shapes3D (see Table 3).
>
> Table C-1: Influence of the two inductive biases on MPI3D. For EncDec w/o Diff, we replace the diffusion model with a decoder while cross-attention is preserved. For EncDiff w/ AdaGN, we replace the cross-attention with AdaGN.
>
>
> | Methods  | Factor VAE score  | DCI |
> | :-----:| :----: | :----: |
> | EncDiff  w/o diff |0.355 ± 0.075 | 0.143 ± 0.038 |
> | EncDiff w/ AdaGN| 0.592 ± 0.111 |  0.268 ± 0.062 |
> | EncDiff | 0.872 ± 0.049 | 0.685 ± 0.044 |
>
> Table C-2: Ablation study on the two design alternatives on obtaining the token representations.
> | Methods  | Factor VAE score | DCI |
> | :-----:| :----: | :----: |
> | EncDiff-V|0.863 ± 0.075 | 0.629 ± 0.047 |
> | EncDiff | 0.872 ± 0.049 | 0.685 ± 0.044 |
>
>
>
> **Q2**: The model does not compare itself to Diff-AE[24] because it does not explicitly disentangle the data. However, to show improvement, it is essential to compare the models to see if they are better since they are similar in many aspects. Diff-AE does show disentanglement qualities in the paper. In addition, a comparison of high-quality Diff-AE and the suggested method could be interesting.
>
> **A2**: Thank you for your helpful suggestion. We followed your suggestions and evaluated Diff-AE on Shapes3D as shown in Table D below.  Consistent with the trends on CelebA (see Table 2 in the manuscript),  our method outperforms Diff-AE with a large margin on Shapes3D.
>
> Table D: Performance comparison with Diff-AE on Shapes3D.
> | Methods  | Factor VAE score | DCI |
> | :-----:| :----: | :----: |
> | Diff-AE| 0.1744 | 0.0653  |
> | EncDiff | 0.872 |  0.685  |
>
> **Q3**: The resource comparison between the method and other competitive methods is unclear and lacks empirical results.
>
> **A3**: Besides the computational complexity comparison in Table 6 in our manuscript, we will add more comparisons in Table E below. In particular, we additionally added the computational complexity of the VAEs and GANs. The computational complexity and inference time of VAEs and GANs still have strength, while diffusion has better generation and disentangling ability (see Table 2). When compared with other diffusion-based models, EncDiff has better generation and disentangling ability but less computational cost and inference time.
>
> Table E: Computational complexity comparison.
> | Models |  Params (M) | FLOPs (M) | Time (s) |
> | :-----:| :----: |  :----: |:----: |
> | FactorVAE | 11.9 | 892.1 | < 1|
> | BetaTCVAE | 7.9 |  542.1 | < 1|
> | DisCo | 12 |  907.2 | < 1|
>
> **Q4**: It is unclear to me if section 3.1 is claimed to be the background or contribution of the paper?
>
> **A4**: We will revise it for better clarity.  Section 3.1 is the contribution of the paper which introduces our overall framework.
>
> **Q5**: In Eq.4, is there a typo maybe?
>
> **A5**: Thank you for pointing it out. It is a typo here and we will revise it.

---

> > ### Comment · Reviewer_Mm6Q · 2024-08-09
> > **Response**
> >
> > Thank you for your kind response.
> > My concerns have been addressed, and I will raise my score.

---

> > > ### Author Response · Authors · 2024-08-09
> > >
> > > Thank you very much for your great effort and valuable feedback!

---

### Official Review · Reviewer_rNx8 · 2024-07-08

**Soundness:** 3
**Presentation:** 3
**Contribution:** 2
**Rating:** 5
**Confidence:** 4

**Summary:**

This paper proposes a representation learning method which employs an encoder as part of a latent diffusion model. During training the encoder takes the target clean image and encodes it into a compressed representation vector. This vector is then used to condition the denoising UNet as it tries to denoise the noisy observation. The conditioning is done with cross attention where queries come from the UNet inner layer activations and key and values come from the conditioning vector.
The method is trained end to end on some simple datasets and is shown to learn disentangled representations, unsupervised. Results are compared to some existing disentanglement methods with favourable results. It is also qualitatively shown that at least on simple datasets the method works well - each factor in the learned representation learns a different source of variability in the data - colour, orientation etc.

**Strengths:**

This model delves into a relatively underexplored side of diffusion models and while not particularly original is an interesting specific and simple (in a good way) combination of existing methods. See below for some reservations about the originality (though not a major decision factor here).

The paper supports, on quite a small scale, most of its claims. The experiments are mostly well run and analysis and ablation is adequate. I particularly liked the attention visualization figures.

The paper is well structured, figures are clear and all in all well presented. I found the language and writing a bit dense however.

**Weaknesses:**

All in all this is a nice paper which suffers from a few (sometimes major) weaknesses.

Disentanglement is a thorny subject when it comes to larger scale experiments. Not just technically, but because when data is more complex the actual definition of what are the disentangled factors becomes blurrier and blurrier. In that sense the paper is inherently in trouble - because the paper focuses so much on disentanglement there is almost no point of me criticising the small scale of experiments, but I will still do it. Using such small datasets really does take away from the potential strength of the paper - these toy-ish datasets are good to get a general idea of how a method works, but I don't think that it can be the final experimental set in today's standards. Gains on "disentanglement" metrics don't mean much in my opinion, and the differences between different model on such simple data is, I think, negligible.

That being said, even for this specific problem setup, there are still issues at hand - the use of "latent" diffusion, as opposed to pixel diffusion, is both surprising, not explained in the paper and not analyzed properly. I would expect to see results directly on pixels, taking into account the potential of the LD encoder to perform much of the work of disentanglement. No appropriate experiments is shown in the paper, and the LD use is taken for granted.

Finally, there are some missing works and baselines that at least should have been discussed, if not compared to in the paper: DIPVAE (Variational inference of disentangled latent concepts from unlabeled observations. 2018), and more recently SODA (https://arxiv.org/abs/2311.17901)

**Questions:**

See above.

**Limitations:**

These are mostly discussed in the paper.

---

> ### Author Rebuttal · Authors · 2024-08-07
>
> Thank you very much for your constructive suggestions, and the appreciation of the method, adequate experiments, attention visualization, and paper structure. We understand your concerns and we aim to move a small step to advance this field and inspire more future works. Please find our detailed responses below.
>
> **Q1**: Disentanglement is a thorny subject when it comes to larger scale experiments. Not just technically, but because when data is more complex the actual definition of what are the disentangled factors becomes blurrier and blurrier. In that sense the paper is inherently in trouble - because the paper focuses so much on disentanglement there is almost no point of me criticising the small scale of experiments, but I will still do it. Using such small datasets really does take away from the potential strength of the paper - these toy-ish datasets are good to get a general idea of how a method works, but I don't think that it can be the final experimental set in today's standards. Gains on "disentanglement" metrics don't mean much in my opinion, and the differences between different model on such simple data is, I think, negligible.
>
> **A1**: Thank you for your feedback. As commented by you, disentanglement is a thorny subject when it comes to larger scale experiments because the actual definition of what are the disentangled factors becomes blurrier and blurrier. We agree that disentanglement is still at the dawn of its development. It is still a very challenging but valuable field, with great potential to reshape the technique revolution. We endeavor to move a step forward to expand the boundary, even though we are still far from the terminus.
>
> In this paper, we introduce a new perspective and framework, demonstrating that diffusion models with cross attention can serve as powerful inductive bias to facilitate the learning of disentangled representations. **The experimental results (see results in Table 2) on the real-world dataset CelebA, which contains 10,177 number of identities with 202,599 number of face images, reveals our method’s potential to extend to more complex datasets.**  Compared with the state-of-the-art approach DisDiff [37], our EncDiff significantly improves the disentanglement performance from 0.305 to 0.638 in terms of TAD on CelebA.
>
> We made some investigation during the rebuttal on some other real world data. Based on the key idea/perspective of this work, we designed a similar framework, which inherits the exploration of the inductive biases of time-varying bottleneck and cross attention of diffusion model, to study the disentanglement of semantics from images. We show the framework and results in **our rebuttal pdf file**. The target is to disentangle concepts or properties (color, long-hair, big-eared) from the inverted objects (white dog). If the concept were disentangled, we can combine the concepts from different instances to create new objects (e.g., white big ear dog: combine white dog and big-eared dog). Figure 2,3,5 demonstrated that properties (color, long-hair, big-eared) are learned in our framework where the concepts are correctly swapped.
>
> We will conduct more deeper study on complex datasets in future.
>
> We hope our work will inspire further investigations on diffusion for disentanglement to address the more sophisticated data analysis and understanding.
>
>
> **Q2**: I would expect to see results directly on pixels, taking into account the potential of the LD encoder to perform much of the work of disentanglement.
>
> **A2**: Thanks for your insightful suggestion. As described in Section 3.2.1 line 156, our analysis also holds in pixel space. Following your suggestion, we trained EncDiff directly on pixel space on the Shapes3D dataset, and show the disentanglement results in Table B below. Our framework in pixel space still achieves excellent disentanglement performance, demonstrating the LD encoder is not the key for achieving disentanglement. We will conduct experiments on other datasets and add more analysis in our revision.
>
> Table B: Disentanglement performance of our EncDiff in pixel space (EncDiff pixel) and latent space (EncDiff) in terms of factor VAE score, and DCI (both the higher the better), evaluated on the Shapes3D dataset.
> | Models | factor VAE score | DCI |
> | :-----:| :----: | :----: |
> | EncDiff pixel |1.0 ± 0. | 0.981 ± 0.015 |
> | EncDiff |0.999 ± 0.000 | 0.969 ± 0.030|
>
> **Q3**: There are some missing works and baselines that at least should have been discussed, if not compared to in the paper.
>
> **A3**: Thank you very much for pointing out the two related works. We will add the following discussion in the related work section: DIP-VAE introduces a regularizer on the expectation of the approximate posterior over observed data, by matching the moments of the distributions of latents. The recent work SODA leverages a diffusion model for representation learning, revealing its capability of capturing visual semantics. In this work, we analyze and identify two valuable inductive biases of diffusion that promote the disentangled representation learning in our framework.

---

> > ### Comment · Reviewer_rNx8 · 2024-08-12
> > **Thank you**
> >
> > Thank you taking the time to answer my concerns.
> >
> > Beyond the basic limitations of disentagnlement papers most of my concerns have been answered and I am raising my score.

---

> ### Author Response · Authors · 2024-08-12
>
> Thank you very much for your great efforts in reviewing our paper and responses. We appreciate your thoughtful and constructive feedback  and will incorporate the suggestions into our revision.

---

### Official Review · Reviewer_HVuH · 2024-07-25

**Soundness:** 3
**Presentation:** 3
**Contribution:** 4
**Rating:** 7
**Confidence:** 3

**Summary:**

The paper proposes a novel method that utilizes a concept-extracting image encoder and the cross-attention mechanism in conditional diffusion models for achieving the learning of disentangled representations. Comprehensive experiments, visualizations and ablation studies confirm the effectiveness of the proposed method.

**Strengths:**

I find this paper a strong submission.

*Novelty*. To the best of my knowledge, utilizing cross-attention with features from an image encoder to achieve feature disentanglement is a novel approach.

*Presentation*. The paper is overall well-written and organized. The figures for framework/concept demonstration are also quite clear.

*Good intuition and solid experiment*. The proposed method is well-motivated  and the paper demonstrates strong empirical performance. Furthermore, the empirical observations validate the functionality of each component in the proposed method, making the paper more sound.

**Weaknesses:**

I don't see any major flaws in the paper.

*Typos and writing related*.
*  Line 174, it would be better to briefly describe why this information bottleneck promotes disentanglement instead of just cite other papers.
* Line 282, "utilizing reconstruction l2 loss is used to optimize the entire network." utilizing and used are redundant.
* In 4.1, Implementation Details does not mention details about the diffusion model, readers can mistakenly think that diffusion models are not trained.
* I don't understand the first ablation study "Using Diffusion as Decoder or Not". The designed experiment seems to remove the upper half of a U-Net model, why does this provide evidence about the importance of diffusion?
* Table 6 only list computational complexity of diffusion-based method, it would be good to also include other baseline methods compared in Table 1.

**Questions:**

* The current method is trained end-to-end and the authors mention that achieving disentanglement in complex data is still hard. Is it possible to fine-tune existing pre-trained LDM using the proposed approach for achieving better disentanglement in complex data?
* In the third ablation study, why does scalar-valued perform better? It seems that the vector-based can potentially extract more information and hence perform better? Can the author further elaborate here?

**Limitations:**

Yes

---

> ### Author Rebuttal · Authors · 2024-08-07
>
> Thank you very much for your appreciation and recognition of our work regarding the novelty, writing, and strong performance. We will incorporate your helpful suggestions into our revision.
>
> **Q1**: Typos and writing related.
>
> **A1**: Thank you very much  for your helpful suggestions. We will clarify/rewrite those in our revision.
>
> **Line 174**: We will add more explanation as: The information bottleneck promotes disentanglement by forcing the model to efficiently compress the input data into a limited latent space. This constraint encourages the model to represent distinct and independent features of the input in separate latent dimensions. As a result, each latent variable tends to capture a unique aspect of the data, leading to disentangled representations where different latent variables correspond to different generative factors.
>
> **Line 282**: We will remove “is used”.
>
> **Details in 4.1**: We will move such crucial details from Appendix D to Section 4.1.
>
>  **About ablation on Using Diffusion as Decoder or Not**: We will make it clearer in our revision. We designed a variant (EncDec w/o Diff) of EncDiff to have  an autoencoder-like structure, by reusing the image encoder as encoder and the lower half of the U-Net structure as decoder for reconstruction. In contrast to  EncDiff, we discard the multiple step diffusion process and only run once feedforward inference to get the reconstruction.  If the autoencoder’s performance drops significantly, this will provide evidence about the importance of the diffusion process instead of the U-Net architecture.
>
> **Computational complexity on more methods**: The computational complexity of the VAE-based  and GAN-based methods are also listed as follows. The computational complexity and inference time VAEs and GANs still have strength, but diffusion has much better generation and disentangling ability. Among the diffusion models, our EncDiff has better generation and disentangling ability but less computational cost and inference time.
>
> Table A: Computational complexity comparison.
> | Models |  Params (M) | FLOPs (M) | Time (s) |
> | :-----:| :----: |  :----: |:----: |
> | FactorVAE | 11.9 | 892.1 | < 1|
> | BetaTCVAE | 7.9 |  542.1 | < 1|
> | DisCo | 12 |  907.2 | < 1|
>
>
> **Q2**: Is it possible to fine-tune existing pre-trained LDM using the proposed approach for achieving better disentanglement in complex data?
>
> **A2**: Thank you for the helpful suggestion. We agree and believe that leveraging the pre-trained LDM would ease the disentanglement in complex data. Due to limited rebuttal time for implementation, we will add the studies in our revision.
>
>
> **Q3**: In the third ablation study, why does scalar-valued perform better? It seems that the vector-based can potentially extract more information and hence perform better? Can the author further elaborate here?
>
> **A3**: Consistent with your understanding, we think that the vector-based representation potentially extracts more information and hence enforces a looser bottleneck than scalar-valued representation.   Please note that the more information encoded, there is a higher probability that the encoded information is correlated, which is contradictory for disentanglement. Therefore, the tighter bottleneck from scalar-valued representation leads to (a slightly) better performance. We will add more explanation in the revision.

---

> > ### Comment · Reviewer_HVuH · 2024-08-13
> > **Response to rebuttal**
> >
> > I thank the authors for the rebuttal.
> >
> > After reading it along with other reviews, I'll keep my original score since I believe this is a strong submission. In the meantime, due to the somewhat simplified experimental setting, I'll not raise the score further.

---

> > > ### Author Response · Authors · 2024-08-13
> > >
> > > Thank you very much for your great efforts in reviewing our paper and responses, and the recognition of our work! We appreciate your valuable feedback and will incorporate these good suggestions into our revision.

---

### Author Rebuttal · Authors · 2024-08-07

Dear Reviewers,

We sincerely appreciate the time and effort you have invested in reviewing our manuscript. Your constructive feedback has been instrumental in identifying areas for improvement, and we are grateful for your positive feedback on paper novelty (Reviewer HVuH, Mm6Q), good intuition (Reviewer HVuH, Mm6Q), strong (Reviewer HVuH, Mm6Q) performance and adequate ablation (Reviewer HVuH, rNx8), and paper presentation (Reviewer HVuH, rNx8, Mm6Q).

We have carefully considered each of your comments and suggestions, and below, we provide detailed responses to address the concerns raised. We believe that incorporating your valuable insights will significantly enhance the quality and clarity of our paper.

We are committed to making the necessary revisions and are eager to engage in further discussions. Your additional questions or concerns are most welcome, as they will help us refine our work to meet the high standards of the conference.

Thank you once again for your invaluable input.

Best regards,

All authors

---

### Decision · Program_Chairs · 2024-09-25

**Decision:**

Accept (spotlight)

**Comment:**

The paper presents a novel approach to learning disentangled representations. There is a consensus in the reviews that this paper brings a valuable contribution to the field. Some concerns where raised regarding the emprical evaluation, specifically regarding the number of datasets and the selection of models used for comparison. This is a common concern with diesntanglement papers, particularly when they are contrasted to supervised learning papers. In this context, however, developing empirical evaluation with the similar scale (data sets) is significantly more challenging.  Overall, the authors present a solid and broad evaluation and have constructively adopted suggestions from the reviews.